

# Spatio-temporal consistent bias pattern detection on MIROC5 and FGOALS-g2

Bo Cao[1], Ying Zhao[1], and Ziheng Zhou[1]

[1]Department of Computer Science and Technology, Tsinghua University, Beijing, China

**Correspondence:** Ying Zhao (yingz@tsinghua.edu.cn)

**Abstract.** Building climate models is a typical means of studying the dynamics of the climate system and assessing the impacts of climate change. However, model-related biases are common in existing climate models, such as the double ITCZ bias in most CMIP5 models. Recent studies suggest that biases showing distinct spatio-temporal characteristics may involve different mechanisms and sources in climate models. More dedicated studies on bias patterns is important not only for improving model
performance, but also for helping modelers to better understand the climate system. In this paper, we focus on detecting *spatio-temporal consistent* bias patterns from climate model outputs. A spatio-temporal pattern is a bias pattern that is present in contiguous space with significant and coherent biases in continuous time periods. These patterns are ideal for revealing regional and seasonal characteristics of biases and worth further investigation by modelers. Due to the high computation cost, most of the existing analysis methods can only detect bias patterns that are either spatial consistent or temporal consistent,
but not both at the same time. We proposed a bottom-up algorithm to tackle this problem. The proposed method first detects regions showing significant and consistent biases at each time slot and then merges them iteratively to form bias instances. The resulting bias instances are further clustered into different families to depict corresponding spatio-temporal consistent bias patterns. The experiments on both MIROC5 and FGOALS-g2 precipitation outputs show that the proposed approach can detect some important bias patterns that are consistent with previous studies and can produce other interesting findings. Modelers can
adopt the proposed method as an exploratory tool to develop insights for bias correction and model improvement.

## 1 Introduction

Modern coupled atmosphere-ocean general circulation models (GCMs) are the typical means of studying the dynamics of the climate system and assessing the impact of climate change (Hori and Ueda, 2006; Dufresne and Bony, 2008; Semenov and
Stratonovitch, 2010). However, model-related uncertainties and biases are commonly present in climate models in the Coupled Model Intercomparison Project Phase 5 (CMIP5), such as double-intertropical convergence zone (ITCZ) bias (DIB) (Oueslati and Bellon, 2015) and excessive equatorial Pacific cold tongue bias (Li and Xie, 2014). Recent studies suggest that the bias patterns that exhibit distinct spatio-temporal characteristics may involve different mechanisms and have different sources in



GCMs (Adam et al., 2018), which could play an important role in improving GCMs and demand more advanced bias pattern detection methods rather than traditional bias analysis approaches (e.g., predefined climate indices or empirical orthogonal functions (EOFs) (North et al., 1982)). In particular, we focus on detecting the bias patterns in climate model outputs present in contiguous space that are significant and coherent during continuous time periods, which we call *spatio-temporal consistent*
bias patterns in this paper. These patterns are ideal for revealing regional and seasonal characteristics of biases and worth further investigation by modelers. These patterns are also local in nature, i.e., only present in local areas and time periods, which are often missed by global analysis methods such as EOFs.

Current bias analysis approaches are not sufficient for detecting spatio-temporal consistent bias patterns. Initially, biases in the format of time series of grid residuals can be obtained by comparing climate model outputs with observed climate data.
Among current analysis methods on those residual time series, empirical orthogonal functions (EOFs) (North et al., 1982), rotated EOFs (Richman, 1986), single value decomposition (SVD) (Golub and Reinsch, 1971) and other variants of principle component analysis methods are the most popular ones. The main idea is to decompose spatio-temporal data into spatial patterns and associated time indices that capture the major variances in the data. Model outputs can be evaluated by comparing their EOF spatial patterns with those of the observational data or by applying the EOF analysis to residual time series directly.
Chen et al. (2000) used EOFs to analyze the systematic bias of an ocean-atmosphere coupled model and designed a statistical method to correct the model output based on the extracted bias components. Ahmadalipour et al. (2017) performed EOF analysis and SVD on 20 different global climate models in their work to capture the reliability and nature of the particular model at regional scale. Davini and Cagnazzo (2014) used the first empirical orthogonal function to uncover why some state-of-the-art climate models have biased NAOs. Li and Xie (2014) performed an intermodel EOF analysis of tropical Pacific
precipitation coming from 18 coupled GCMs of CMIP5 and all available AMIP simulations. Through this study, they found that the excessive equatorial Pacific cold tongue and double-intertropical convergence zone are the most prominent errors of the current generation of CGCMs. Since the goal of EOFs is to explain the largest variance present in biases over the entire time period, the extracted EOF modes does not necessarily show spatio-temporal consistency.

Knowledge Discovery through Data mining (KDD) methods (Cios et al., 1998; Miller and Han, 2009) such as clustering
(Rui and Wunsch, 2005) are also widely used for analyzing climate model outputs. Given a spatio-temporal field, grids can be classified into different clusters according to their similarity in climatic characteristics. By comparing the clusters found in climate model outputs and the ones found in observations, we can figure out the ability of a model to simulate some specific processes. Hoffman et al. (2005) used a clustering technique called multivariate spatio-temporal clustering (MSTC) to discover recurring climate regimes from model outputs and establish a basis to compare different models. Gómez-Navarro
et al. (2018) used clustering to define a number of non-overlapping bias regions and then corrected the simulated precipitation for these regions separately. More clustering analysis has been done on observed climate data as well. Clustering analysis on temperature and precipitation was performed to discover climate zones in Turkey (Unal et al., 2003), Italy (Di Giuseppe et al., 2013), and China (Zhang and Yan, 2014). Fountalis (2016) proposed a region-growing method called $\delta$-Maps to identify potentially overlapping and spatially contiguous domains from climate data. The method first finds several seeds with high
local homogeneity and then iteratively expands and merges regions to construct homogeneous domains.





Clustering analysis methods can also be used to find interesting patterns with temporal characteristics in climate data. Berrocal (2016) clustered the errors of regional climate models along the time dimension, aiming to identify time periods that require a further in-depth examination. Their work is restricted to a local area and based on the whole region. Runge et al. (2015) and Pnevmatikakis et al. (2016) tried to consider temporal characteristics by matrix factorization. They first used

matrix factorization to reduce the length of time series and then clustered the new series. Subsequence clustering on time series is another important way to extract temporal patterns. The most important concept in time series subsequence clustering is "shapelet", which is defined as a small, local pattern in a time series that can be used to distinguish a class from the others. Zakaria et al. (2012) designed an unsupervised algorithm to find such shapelets. However, their algorithm can extract only one shapelet from a time series with no spatial information. To conclude, most of the existing clustering analysis methods focus

on either spatial or temporal characteristics of climate data, but not both due to the high computational cost. In this paper, we propose a new approach to detect spatio-temporal consistent bias patterns from climate model outputs, which considers spatial and temporal characteristics simultaneously.

The proposed approach can be separated into four steps: Firstly, regions showing significant bias are identified at each time slot. Secondly, regions sharing enough common objects at adjacent time slots are merged to form bias instances. Thirdly, result-

ing bias instances are clustered into bias families based on their spatial and temporal characteristics. Finally, characteristics of detected bias families are visualized. We applied the approach to analyze precipitation simulation outputs in historical runs of two CMIP5 models, MIROC5 developed by Japan and FGOALS-g2 developed by China. The top 20 bias families were mainly analyzed and the results showed that our approach can produce many important bias patterns consistent with previous studies as well as novel bias patterns. For example, our approach can capture the individual spatio-temporal characteristics of biases

in MIROC5 and FGOALS-g2 in terms of the double ITCZ bias (Oueslati and Bellon, 2015). The proposed approach can also detect underestimated precipitation over the Indian continental and overestimated precipitation over the adjacent ocean during the summer moonsoon in FGOALS-g2 (HUANG Xin, 2019). In addition to these widely studied bias patterns, the proposed approach depicts other interesting bias patterns for both CMIP5 models as well.

As an exploratory tool, the proposed method can discover spatio-temporal consistent bias patterns from climate model simu-

lation outputs. Modelers can thus focus on the corresponding area and time period, study the source of the bias, understand the internal mechanism and improve the model. The resulting bias families can also serve as vertices in network analysis methods, such as Runge et al. (2015) and Fountalis (2016). By performing network analysis, relations between different meteorological factors and different areas can be extracted. We may answer questions such as how a bias spreads in space and in time.

## 2   Methods

Suppose observational data and climate model outputs are all gridded data, with each grid point $g_i$ represented by its latitude $g_i.lat$ and longitude $g_i.lon$. Suppose $\mathbf{R}$ is a 3-dimensional residual matrix calculated by subtracting time series of gridded observational data from climate model outputs, whose dimensions are latitude, longitude, and time, respectively. $\mathbf{R}[:,:,t]$





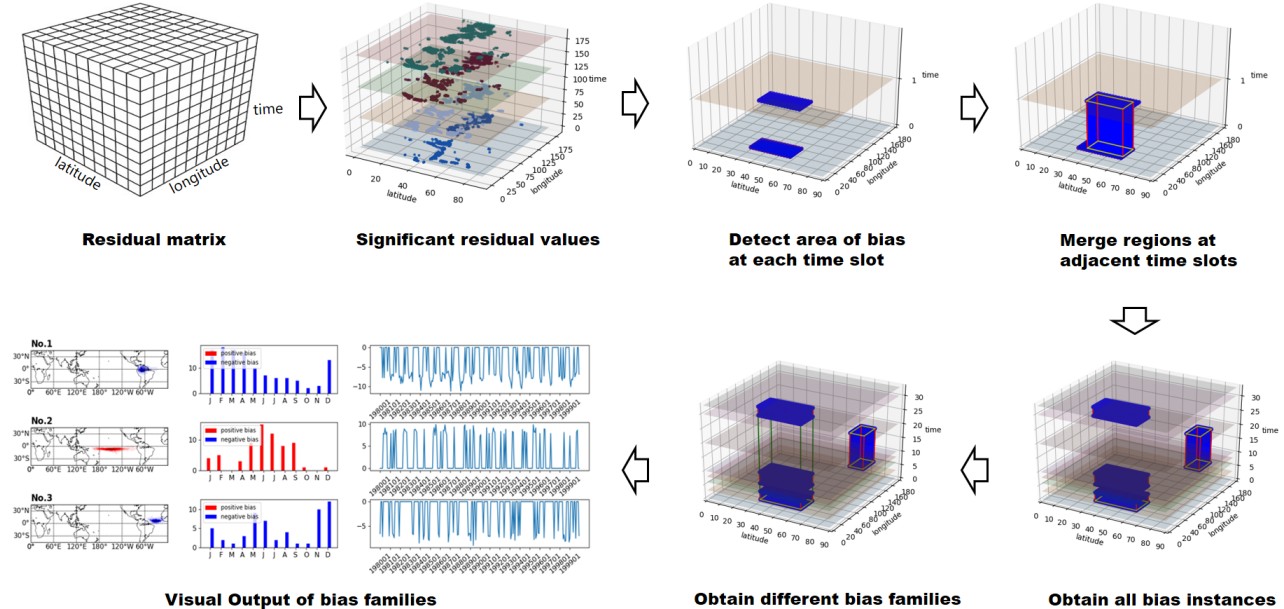

**Figure 1.** Framework of detecting spatio-temporal consistent bias patterns in climate model outputs. Given a residual matrix shown in (a), the proposed approach has four steps: Firstly, (b) regions showing significant bias are identified at each time slot. Secondly, (c-d) regions sharing enough common objects at adjacent time slots are merged to form bias instances. Thirdly, (e) resulting bias instances are clustered into bias families based on their spatial and temporal characteristics. Finally, (f) characteristics of detected bias families are visualized.

denotes a 2-dimensional matrix, representing residuals on each grid point at time $t$. We propose a method that detects spatio-temporal consistent bias patterns in $\mathbf{R}$, whose framework is given in Figure 1.

## 2.1 Bias instances

Suppose $\mathbf{R}_{bi}$ is a subset of the residual matrix within a certain time period in a specific region, indexed by $bi = \langle region, tStart, tEnd \rangle$,

5  where $bi.region$ is a set of $n$ spatially contiguous grid points $\langle g_1, \ldots, g_n \rangle$, $bi.tStart$ and $bi.tEnd$ indicate the time period that $\mathbf{R}_{bi}$ spans. We call $\mathbf{R}_{bi}$ is a **bias instance**, if $\mathbf{R}_{bi}$ satisfies the following conditions:

(1). The number of grid points in $bi$ must be larger than a user defined threshold $\delta_a$, i.e., $n \geq \delta_a$.

(2). The bias on all grid points of $bi$ must be *significant* in $\langle bi.tStart, bi.tEnd \rangle$ by a statistical test with a user-defined significant level $\gamma_r$.

10  (3). The bias on all grid points of $bi$ must be *coherent* in $\langle bi.tStart, bi.tEnd \rangle$, i.e., residuals are all positive or all negative.

(4). Lengthening the time period or expanding the region of $bi$ will violate condition (2) or condition (3).

We propose an algorithm to detect all bias instances in $\mathbf{R}$ by first identifying spatially contiguous regions with coherent biases at each time slot and then merging such regions across adjacent time slots with sufficient large common area. Given three inputs $\mathbf{R}$, $\delta_a$, and $\gamma_r$, the algorithm can detect all bias instances in $\mathbf{R}$ in the following six steps:



(1). For each time slot $t$, initialize an undirected graph $G_t(V, E)$ with each grid point with significant bias as nodes and empty edges;

(2). Add edges in $G_t$ between two spatially adjacent nodes with coherent biases;

(3). Find all connected components with at least $\delta_a$ nodes of $G_t$;

(4). Convert each connected component $c_i$ to a bias instance index $bi_i$: $bi_i.region = c_i, bi_i.tStart = bi_i.tEnd = t$;

(5). Merge two bias instances indexed by $bi_i$ and $bi_j$ in adjacent time slots, if their overlapped area contains at least $\delta_a$ grid points;

(6). Return all detected bias instances.

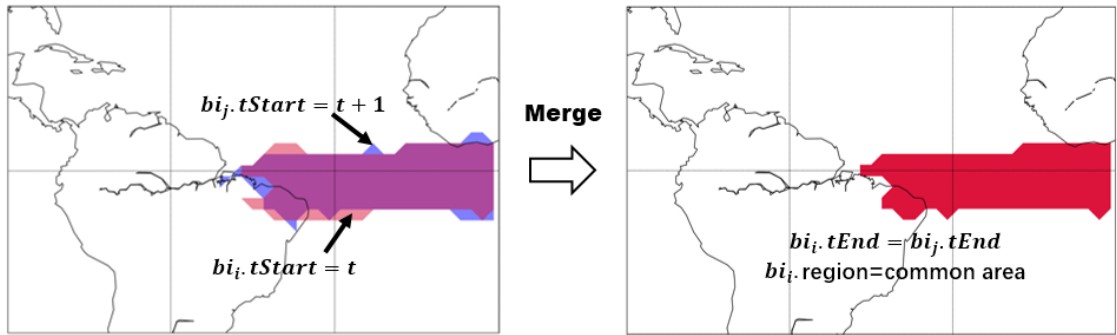

**Figure 2.** Merge bias instances indexed by $bi_i$ and $bi_j$ in adjacent time slots if their overlapped area contains at least $\delta_a$ grid points. The merged bias instance is indexed by $bi_i$ with a retained overlapped area and an extended duration.

Step (5) adopts a bottom-up strategy to merge bias instances across adjacent time slots. For a bias instance indexed by $bi_i$, if we can find a bias instance indexed by $bi_j$ whose $bi_j.tStart = bi_i.tEnd + 1$ and their overlapped area contains at least $\delta_a$ grid points, $bi_i$ and $bi_j$ will be merged. This process is shown in Figure 2. Notice that, the overlapped area of $b_i.region$ and $b_j.region$ may consist of several separate regions. The region with most grid points will be considered as their common area.

## 2.2 Bias families

After identifying all bias instances, we want to discover bias instances that occur in the same region and have similar temporal trends. These instances will be grouped into the same family in this step. Two bias instances indexed by $bi_i$ and $bi_j$ are said to belong to the same **bias family** if they satisfy the following two conditions:

(a)

$$\text{card}(\text{Overlap}(bi_i.region, bi_j.region)) \geq \delta_r * \max(\text{card}(bi_i.region), \text{card}(bi_j.region)), \tag{1}$$

where $\text{card}()$ returns the cardinal number of a set, $\text{Overlap}()$ calculates the overlapped region of two regions, and $\delta_r$ is a user defined parameter in range $[0, 1]$.




(b).

$$\frac{1}{\mathrm{card}(bi_i.center) + \mathrm{card}(bi_j.center)}\mathrm{DTWDistance}(bi_i.center, bi_j.center) \leq \delta_f, \tag{2}$$

where $\mathrm{DTWDistance}()$ is the distance between two time series given by the dynamic time warping (DTW) technique (Senin, 2008) normalized by the length of two time series, and $bi_i.center$ and $bi_j.center$ are two time series of average biases in bias instances indexed by $bi_i$ and $bi_j$, respectively. Given a bias instance indexed by $bi_i$, the time series of average bias within $bi_i.region$ during $[bi_i.tStart, bi_j.tEnd]$ is defined as:

$$\forall t \in [bi_i.tStart, bi_j.tEnd], \ \ bi_i.center[t] = \frac{1}{\mathrm{card}(bi_i.region)}\sum_{g \in bi_i.region}\mathbf{R}[g.lat, g.lon, t] \tag{3}$$

.

$\delta_f$ can be determined in the following way: Normalized DTW distances of all bias instance pairs are calculated, whose mean and standard deviation are $\mu_d$ and $\sigma_d$, respectively. $\delta_f$ is set to $\mu_d - \gamma_d \times \sigma_d$, where $\gamma_d$ is a user defined parameter (1 by default). Larger $\gamma_d$ means stricter similarity requirement among bias instances in a family.

The algorithm for detecting bias families takes three inputs: (1) previously detected bias instances $bis$ in $\mathbf{R}$; (2) $\delta_r$ minimum ratio that two bias instances should overlap; (3) $\gamma_d$ to determine the maximum normalized DTW distance threshold $\delta_f$. There are four steps in the algorithm:

(1). Initialize an undirected graph $G(V, E)$ with $bis$ as nodes and empty edges;

(2). Add an edge $e_{ij}$ to $E$ if $bi_i$ and $bi_j$ satisfy inequality (2) and inequality (3) simultaneously;

(3). Find all connected components of $G$;

(4). Convert each connected component $c_i$ to a bias family $bf_i$, return all detected bias families.

## 2.3 Characterizing bias families

Our method will output bias families in different figures, sorted by the total duration of all bias instances in a family. A sample output of a bias family is shown in figure 3. The upper part shows the location where the bias family often occurs. The

down-left part shows the distribution of the bias family over different months. Modelers can figure out whether the bias is all year round or just occurs in specific seasons. For example, the bias family in this sample usually occurs in summer. Months can be substituted with other seasonal indicators such as hours in a day. The down-right part shows the bias at different time slots. Non-significant biases are truncated to zero. Modelers can figure out whether the bias family exists in the whole time period or just lasts for a certain time period. Combining these three characteristics, the analysis result can help modelers discover

locations and time periods or seasons requiring a further in-depth examination to improve the model.

## 3  Data and Results

In this section, the proposed bias detection method was applied to simulated global precipitation data to check its effective-ness, compared with the current state-of-the-art spatial clustering method $\delta-$maps for climate data. $\delta-$maps tries to cluster




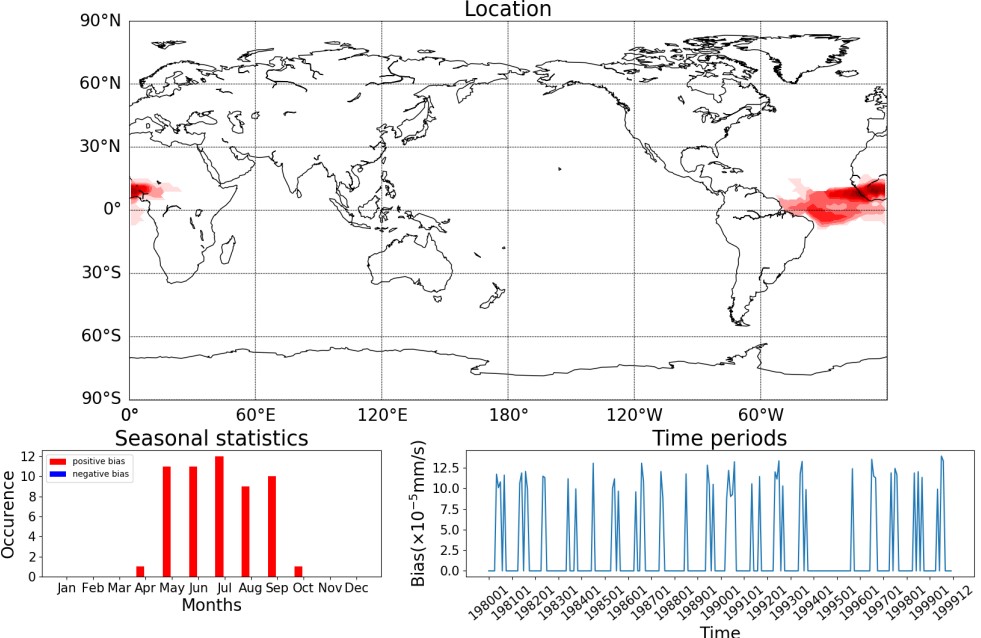

**Figure 3.** Sample output of a bias family detected by the proposed method. The upper part shows the location where the bias family often occurs. The down-left part shows the occurrence distribution of the bias family over various months. The down-right part shows the bias at different time slots. Non-significant biases are truncated to zero.

gridded data basing on the correlation between their associated residual time series. It first chooses some grids having high correlation with surrounding grid points as seeds. Then these grid points will be expanded to form regions and regions with high homogeneity will be merged, which is very similar to hierarchical clustering. This process will stop when homogeneity within clusters achieves a threshold $\delta$. The source code of $\delta-$maps is available on https://github.com/FabriFalasca/delta-MAPS. Default parameters were used in our study. $k$ was set to 16 to find seeds with high homogeneity. $\alpha$ was set to 0.02 to determine the determine the homogeneity threshold $\delta$. $MaxLag$ between time series was set to 12 months. For the proposed method, the parameters were set as follows in this study: $\delta_a = 25$, $\gamma_r = 0.05$, $\delta_r = 0.5$, $\gamma_d = 1$.

### 3.1 Observation and simulation data

GPCP (Global Precipitation Climatology Project) (Adler et al., 2003; Huffman et al., 2009) V2.2 Monthly data from Jan 1980 to Dec 1999 was used as the observational precipitation in this study and was downloaded from ftp://ftp.cdc.noaa.gov/ Datasets/gpcp/v2.2/precip.mon.mean.nc. GPCP is a high quality reanalysis dataset of global precipitation integrating data from rain gauge stations, satellites, and sounding observations. The spatial resolution of GPCP is $2.5°$ latitude $\times 2.5°$ longitude.

The outputs in historical runs of two models MIROC5 and FGOALS-s2 coming from CMIP5 (Coupled Model Intercomparison Project Phase 5) were used as simulation data and they were downloaded from https://esgf-node.llnl.gov/search/cmip5/.



**Table 1.** Simulation Datasets

| Model Name | CMIP Phase | Institute | Spatial Resolution | Ensemble |
|---|---|---|---|---|
| MIROC5 | CMIP5 | AORI (Atmosphere and Ocean Research Institute, the University of Tokyo, Japan), NIES, JAMSTEC (Japan Agency for Marine-Earth Science and Technology, Japan) | $\approx 1.5°$ latitude $\times$ $1.5°$ longitude | r1i1p1 |
| FGOALS-s2 | CMIP5 | IAP (Institute of Atmospheric Physics, China), CAS (Chinese Academy of Sciences, China) | $\approx 1.7°$ latitude $\times$ $2.8°$ longitude | r1i1p1 |

Monthly data from Jan 1980 to Dec 1999 in historical runs.

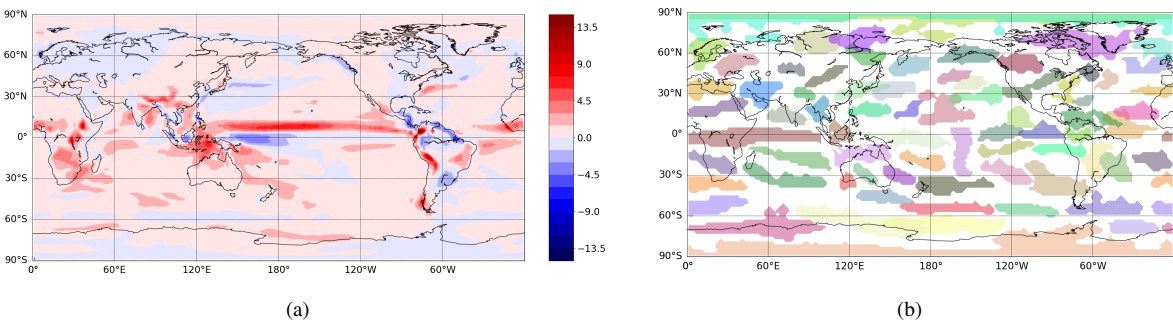

**Figure 4.** (a) Spatial distribution of annual-mean biases in MIROC5 precipitation output ($\times 10^{-5}$mm/s), averaged on 1980-1999. Red represents positive biases and blue represents negative biases. The darker grid points have greater biases. (b) Regions detected from the residual matrix of MIROC5 using $\delta-\text{maps}$. Colors are set arbitrarily to label different regions.

Detailed information of selected models is given in Table 1. Monthly data from Jan 1980 to Dec 1999 were used in our experiment and all data are interpolated to $2°$ latitude $\times$ $2°$ longitude with "area_hi2lores_Wrap" function or "linint2_Wrap" function in NCL (NCAR Command Language Version 6.4.0) (Brown et al., 2012). Grid points with missing values were ignored in this study.

## 3.2 $\delta-\text{maps}$ on MIROC5 precipitation output

Biases in MIROC5 precipitation output are first analyzed with two baseline methods. Figure 4(a) shows the spatial distribution of annual-mean biases, i.e., the 3-dimensional residual matrix is averaged along the time dimension. The resulting 2-dimensional matrix is plotted over the map. Positive residuals are plotted in red and negative residuals are plotted in blue.

The spatial clustering algorithm $\delta-\text{maps}$ is also applied to the residual matrix of MIROC5 precipitation output. The result of $\delta-\text{maps}$ is shown in 4(b), with 72 identified regions plotted with different colors. The resultant regions are not consistent with the ones shown in figure 4(a). Note that spatial clustering algorithms put similar grid points in a cluster, commonly expressed



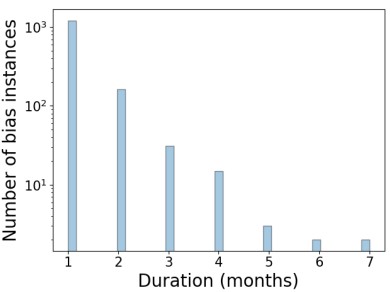

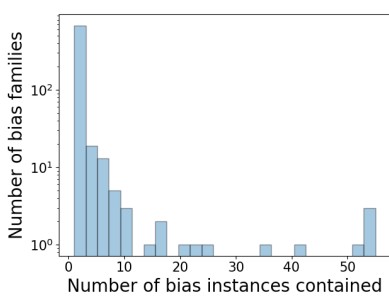

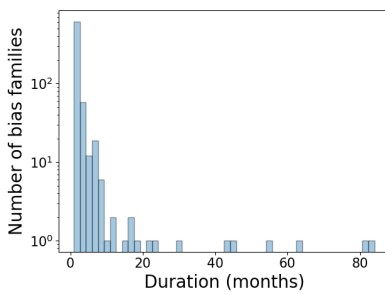

(a) Duration distribution of bias instances      (b) Number of bias instances contained in bias families      (c) Duration distribution of bias families

**Figure 5.** Statistics on detected bias instances and bias families of MIROC5 precipitation output.

in low Euclidean distance or high correlation, here Pearson's correlation coefficient in $\delta-$maps. However, $\delta-$maps measures Pearson's correlation coefficients based on the entire time series. If biases are significant and coherent within a region but only during short time periods, the calculated coefficients will be low. As a result, the significant correlation threshold $\delta$ calculated by $\delta-$maps on the residual matrix is only 0.1043, and the clusters found using this low threshold are mixed with background

noises.

### 3.3 Bias families of MIROC5 precipitation output

There are 1423 detected bias instances from MIROC5 precipitation output and they are classified into 725 families. Figure 5(a) shows the duration distribution of those bias instances. Most bias instances last for less than 4 months. Figure 5(b) shows the size distribution of detected bias families in terms of the number of bias instances contained. Large bias families mean that

some biases occur repeatedly in the same region and that clustering bias instances into bias families is meaningful. Figure 5(c) shows the duration distribution of detected bias families, where the duration of a bias family is defined as the summation of durations of all bias instances in this family. The most frequent bias family lasts for only 82 months, about 34% of the entire time period (240 months). In longer time series, the percentage will be even lower, which indicates that analysis or similarity calculations based on the entire time series could be misleading.

As mentioned in section 2.3, bias families are outputted in decreasing order of their durations. We mainly analyzed top 20 bias families detected in MIROC5 precipitation output. The locations of complete top 20 bias families and their distributions over various months are give in Figure 6 and Figure 7, respectively. The detected bias families are consistent with the ones in Figure 4(a), but with temporal details. As can be seen in Figure 7, some bias families only occur in specific time periods. For example, bias family No.3 often occurs in boreal winter (Watanabe et al., 2010), while bias family No.4 often occurs in

summer (Siongco et al., 2015). Figure 7 also shows bias regions that cannot be seen in Figure 4. For example, Bias family No.5 and No.20 are two bias families happening at the same region but showing different characteristics. One mainly consists

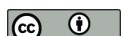



of negative biases while the other mainly consists of positive biases. As a result, this region shows no significant bias after averaging.

Bias family No.2 and bias family No.13 depict the double ITCZ (intertropical convergence zone) bias which has been studied by many researchers (Oueslati and Bellon, 2015; Adam et al., 2018). Actually, MIROC5 can simulates a single ITCZ well, but produces excessive precipitation compared with observation. Previous studies mainly selected two regions to study the double-ITCZ bias: the central Pacific (CP, $130°W - 170°W$) and the eastern Pacific (EP, $80°W - 120°W$). According to the study of Oueslati and Bellon (2015), the precipitation bias of MIROC5 has different patterns in these two areas. The bias in CP area is all year round while the bias in EP area occurs mainly in summer. Bias family No.2 and bias family No.13 can capture this seasonal pattern well. Bias family No.4 depicts another important ITCZ bias in Atlantic area, which was comprehensively studied by Siongco et al. (2015). MIROC5 shows east Atlantic bias which misplaces the precipitation center to the Gulf of Guinea in West Africa.

### 3.4 Bias families of FGOALS-g2 precipitation output

We conducted the same study on FGOALS-g2 precipitation output. The top 20 bias families detected from FGOALS-g2 precipitation output and their distributions over various months are given in Figure 8 and Figure 9, respectively. FGOALS-g2 also shows biases along the equator. Bias family No.1 depicts the spurious ITCZ in the south of the equator in Pacific. In addition, FGOALS-g2 simulates excessive precipitation in the north of the equator in Pacific, shown in bias family No.5, No.12 and No.13. The precipitation in Panama, Columbia and Venezuela is underestimated compared to observation (Ryu and Hayhoe, 2014), shown in bias family No.2 and No.8, similar to bias family No.9 and No.10 in MIROC5. Finally, Bias family No.6 and No.16 depict an important bias corresponding to summer monsoon bias in South Asia (HUANG Xin, 2019).

From the above discussion, we can see that the proposed method can detect spatio-temporal consistent bias patterns effectively. Many of them are important biases having been discovered by modelers manually with careful selection on regions and seasonal cycles. The remaining ones could be studied further to develop more insights for understanding the internal mechanism and leveraging bias correction.

### 4 Summary and Discussion

In this paper, we proposed a method to detect spatio-temporal consistent bias patterns from climate model outputs. The problem that a bias may exist just periodically can be handled well by our method. Through further analysis on such bias patterns in climate model outputs, more insights could be developed for leveraging bias correction, understanding the internal mechanism, and improving climate models. Experiments on precipitation output of two CMIP5 models were conducted to evaluate the effectiveness of our proposed method. Results show that our method can produce several bias patterns consistent with previous research as well as some novel patterns.

In addition, the proposed method can also be used to detect some meteorological processes from observation data. Anomalies in observation data can be analyzed in a similar way using our method. For each grid point, anomaly time series could be





**Figure 6.** Locations of top 20 bias families detected in MIROC5 precipitation output (monthly data from 1980-1999). Red means positive biases and blue means negative biases. Darker grids show significant biases with longer durations.





**Figure 7.** Distribution over months of top 20 bias families detected in MIROC5 precipitation output (monthly data from 1980-1999). Red means positive biases and blue means negative biases. The height of the bar indicates how many times the bias family occurs in the corresponding month.





**Figure 8.** Locations of top 20 bias families detected in FGOALS-g2 precipitation output (monthly data from 1980-1999). Red means the area mainly shows positive bias and blue means the area mainly shows negative bias. Darker grids show significant bias with longer duration.




**Figure 9.** Distribution over months of top 20 bias families detected in FGOALS-g2 precipitation output (monthly data from 1980-1999). Red bar means positive bias and blue bar means negative bias. The height of the bar shows how many times the bias family occurs in the corresponding month.




constructed by subtracting seasonal factors and long term trends from original time series. Then spatio-temporal consistent anomaly patterns can be detected using the proposed method. Anomaly study is a very important field in climate research (Gill and Rasmusson, 1983; Mann et al., 2009; Yeh et al., 2009; Seddon et al., 2016) and these anomaly patterns could correspond to some important meteorological processes, such as El Nino. The resulting bias patterns or anomaly patterns could be used

5 as a basis for network analysis of climate data as well. Network analysis techniques have been applied to climate data in order to depict interaction processes among different functional areas (Donges et al., 2009; Steinhaeuser et al., 2011; Runge et al., 2015). Spatio-temporal consistent bias patterns or anomaly patterns can be treated as a vertex in such networks. By performing network analysis, relations between different meteorological factors and different areas can be extracted. We may answer questions such as how a bias spreads in space and in time or how one meteorological factor affects another one.

10 *Code and data availability.* Code and processed data are available at https://github.com/martincao12/climate_bias_analysis. Source GPCP data can be downloaded from ftp://ftp.cdc.noaa.gov/Datasets/gpcp/v2.2/precip.mon.mean.nc. Source MIROC5 and FGOALS-g2 data can be downloaded from https://esgf-node.llnl.gov/search/cmip5/.

*Author contributions.* B.C. and Y.Z. designed the study. B.C. carried out the study and prepared the manuscript. Z.Z. prepared the data. All authors discussed the results and contributed to editing the manuscript.

15 *Competing interests.* The authors declare that they have no conflict of interest.

*Acknowledgements.* This work is funded by the National Key Research and Development Program of China (Grant No. 2017YFA0604502).



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
