# Peer review of "Spatio-temporal consistent bias pattern detection on MIROC5 and FGOALS-g2"

_Geoscientific Model Development, 2019_

## Referee Comment (RC1) · Anonymous Referee #1 · 17 Jun 2019

This paper presents a new method for detecting and analyzing spatial-temporal consistent bias pattern in climate models. A good introduction is given about the general context, the methods used for climate model evaluation and their limitations in section 1. The method is presented clearly in the method section and the results of the application on precipitation from MIROC5 and FGOALS-g2 compared to the precipitation from GPCP data set are presented in section 3. A very short summary and discussion is found in section 4. The method proposed here is relatively original and could become a useful tool for the analysis of climate data (not only model outputs) through the analysis of anomalies. However, the discussion is very short and lacks comments about the limitations of the method. My main concern is that the authors seem to have compared directly time series from observations and freely evolving climate models. The limita-

tions associated with this issue should be treated with care and properly discussed. Other possible applications of the method as suggested below should be considered in order to propose a more robust analysis of the method and its limitations. The introduction could be slightly improved by adding a couple more information about the context, some perspectives and a few more references. The form of the text is understandable and the number of grammar errors limited although the paper could be improved by having it checked (once again ?) by an English native speaker (which the reviewer is not). I think the paper should undergo major revisions before being accepted.

General comment about the introduction :

Climate models biases spatio-temporal consistency or "stationarity" has important implications for the validity of climate model bias corrections methods (in this regards, see publication by Krinner and Flanner, 2018). This might be worth mentioning in the introduction.

General comment about the method and its application :

1) It is unclear to me whether the matrix of biases (or residuals) for MIROC-5 and FGOALS-g2 were obtained by directly comparing time series of the model to the monthly time series of GPCP data set or to the monthly climatology of GPCP ? In any case, it is challenging or even questionable to directly compare the time series of freely evolving climate models to an observational time series due to the independence between the two time series as freely evolving climate models own their inner variability. Differences for a given month should therefore in any case not being called a "bias". Possibly, the fact that the bias detection algorithm is applied on a long climate period with thresholds on the size of the area and the length of the period to identify "bias families" allows for the method to identify actual biases of the climate model but this issue should be handled with care, more deeply explored and discussed. In my opinion, the application of the method proposed in this paper is more straightforward and easier to justify for the detection of biases in climate simulation (GCM or RCM) nudged towards

meteorological reanalysis and directly compared to time series from observations (discussion of surface climate biases mostly in this case) or for the detection and analysis of climate anomalies as suggested by the authors in their discussion.

2) Why focusing particularly on these two climate models and only on precipitation ? Extending the analysis on other climate models would increase the robustness of the results and the conclusion drawn from them, while applying it to other climate variables (e.g. temperatures) would open possibilities for exploring links between biases among different variables (e.g. temperatures and precipitation) as the method is meant to be a tool for a better understanding of the sources of the climate model biases as the authors mentioned. If an extent of the application and the analysis on other climate models and/or other climate variables, the authors should at least justify why they restrained their analysis to these two climate models and to precipitation.

3) Applied on monthly values as it is in this paper hampers the method from detecting model biases that evolve with the daily cycle (which is often the case for temperatures and precipitation). Would the method proposed in this paper be able to deal with this type of biases ? If this is not explored by a short application of the method on data at a higher time resolution, this should at least be briefly mentioned and discussed.

More particular, minor comments :

Title : If the method is meant to become a useful tool for climate models analysis a more general and "attractive" title should be considered, for example "Spatio-temporal consistent bias pattern detection method : application on MIROC5 and FGOALS-g2 precipitation"

P1 L18 : I would rather recommend the use of AOGCM or CGCM (used but nor defined in P2) acronym when speaking precisely of coupled atmosphere-ocean general circulation models.

P1 L20 : May-be add a more general reference about biases and evaluation of climate

models (e.g. Flato et al., 2013 from IPCC AR5).

P1 L22 : Since it is a widely spread and studied bias, may-be, add one or two more references about the double ITCZ.

P2 L30 : "Highlight" or "evidence" may be more appropriate here than "discover" ?

P3 L13-28 : A brief summary of the method (already present in the abstract) as well as of the results might not relevant at this point in the paper. At this point, readers might just want to know about what they will find in the different sections of the paper.

P4 L8 : Which statistical tests will be available for users to determine significance of the biases ?

P7 L7 : How were the values of the parameters determined ? Did you performed some sensitivity tests ?

P8 L8 "Positive residuals are plotted in red and negative residuals are plotted in blue". I think this sentence should only appear in the caption of the figure.

P9 L13 "In longer time series, the percentage will be even lower, which indicates that analysis or similarity calculations based on the entire series could be misleading" → I am not entirely sure about this statement, and it should in my opinion be explained more clearly if it is a meaningful result.

P9L21 "One mainly consists of negative biases while the other mainly consists of positive biases. As a result...." This result might simply just be the consequence of the model and observational time series being completely independent and disconnected. As already mentioned, I think the authors should deal more carefully with this issue.

References :

Flato, G., Marotzke, J., Abiodun, B., Braconnot, P., Chou, S. C., Collins, W. J., Cox, P., Driouech, F., Emori, S., Eyring, V., et al.: Evaluation of Climate Models. In: Climate Change 2013: The Physical Science Basis. Contribution of Working Group I to the

Fifth Assessment Report of the Intergovernmental Panel on Climate Change, Climate Change 2013, 5, 741–866, 2013.

Krinner, G. and Flanner, M.G., 2018. Striking stationarity of large-scale climate model bias patterns under strong climate change. Proceedings of the National Academy of Sciences, 115(38), pp.9462-9466. https://doi.org/10.1073/pnas.1807912115

———————————————

---

## Short Comment (SC1) · 26 Jul 2019

Thank you for the efforts you have gone to supply the code and data your manuscript depends on. While you seem to have supplied the substance of what is required, the archiving and citation are currently not compliant with GMD policy.

Code on GitHub

GitHub is a great development and distribution platform, but it's not a persistent archive. Even GitHub themselves tell you to use Zenodo for this! Please therefore provide a persistent archive of the exact version of the code that you use. The easiest way to do this is to use the Zenodo GitHub integration. See

https://guides.github.com/activities/citable-code The resulting Zenodo repository will tell you how to cite the resulting DOI. You are welcome to also provide the reference to GitHub, for example as the preferred download location for the current version of the code, but this needs to be in addition to the DOI.

Data citations are insufficiently precise

The comparison model data just points to the ESGF website. It is not clear which model data set was used. For CMIP5 ESGF provides unique data set IDs in the metadata. Please insert the relevant IDs in the code.

For GPCP, the correct citation is given at https://data.nodc.noaa.gov/cgi-bin/iso?id= gov.noaa.ncdc:C00932#. Once again, it's perfectly OK to also include the URL, but the citation is more permanent so should always also be used (you could also put the URL in the citation in the bibliography if you think that's neater).

---

## Referee Comment (RC2) · Juan Antonio Añel (Referee) · 30 Jul 2019

The manuscript presents the description of a method to cluster and classify main regions of bias in 3D grids and its application to the outputs of two climate models. The merit of the work is fair, improved by the fact that the code is shared and licensed under GPLv3. However, there are several shortcomings that need to be addressed.

The proposed methodology is, someway, not new. Clustering analysis has been used to check model performance now for a long time and this is not properly put into context in the Introduction. It is surprising that the work by Yokoi et al. (2011) is not cited at all (https://doi.org/10.1175/2011JAMC2643.1) when moreover uses as examples MIROC5 and FGoals1. However, in the Introduction the authors cite other works

that simply put the focus on regional studies or features, such as the ITCZ. I strongly recommend the authors to perform a better and more complete review of the literature on this topic and improve this section.

Also, the code (see also comment from the executive editor about using ZENODO) includes a README file, but this file does not contain any information about the structure of all the code in the repository. It should explain what each 'program' does and list all the files in the repository. Moreover, the different pieces of code that you share, do not have comments at all. Including commentaries in the code explaining what every set of routines or group of code does, is a basic request in software development and very important when you share it. Therefore, please, include adequate commentaries in the software with explanations. Finally, in the text of the manuscript, when you mention the analysis and the tool to generate Figure 3, include the name of the code/routines that use to do it and make clear to the reader that they can be accessed from the link available in the 'Code and data availability section'.

Other minor issues:

- the current title of the manuscript is misleading. Readers can think that you perform a full detection and study of bias in the two models mentioned. Indeed, your manuscript is about a clustering technique and MIROC and FGOALS are simply used as testbeds without any interpretation. The title should reflect this, maybe something similar to 'An algorithm for spatio-temporal consistent bias pattern detection: a case study using MIROC5 and FGOALS-g2'

- If you want to highlight the relevance of climate models for the study of climate change, probably the papers that you cite in the second line of the Introduction are not the most suitable. Better use Taylor et al. (2012) or IPCC reports.

- When you mention the CMIP5, please, cite the paper by Taylor et al. (2012) (https://doi.org/10.1175/BAMS-D-11-00094.1).

- The last sentence on page 2 is unnecessary, delete it.

- When you state on page 3 (first paragraph) that the computational cost would be too high, please, include extra detail at least on orders of magnitude of how more expensive it would be.

- when you mention the models used, please, provide links and cite the relevant papers with the description of the models.

- the citation and reference to HUANG Xin is capitalized, fix it.

- page 7: you have already explained what CMIP5 stands for in page 1, you do not need to repeat it here.

---

## Author Comment (AC1) · 28 Sep 2019

This paper presents a new method for detecting and analyzing spatial-temporal consistent bias pattern in climate models. A good introduction is given about the general context, the methods used for climate model evaluation and their limitations in section 1. The method is presented clearly in the method section and the results of the application on precipitation from MIROC5 and FGOALS-g2 compared to the precipitation from GPCP data set are presented in section 3. A very short summary and discussion is found in section 4. The method proposed here is relatively original and could become a useful tool for the analysis of climate data (not only model outputs) through the analysis of anomalies. However, the discussion is very short and lacks comments about the limitations of the method. I think the paper should undergo

major revisions before being accepted.

Thank you for taking your time and effort in reviewing our manuscript and we appreciate your insightful comments for improving the manuscript.

General comments:

Q.1-1

My main concern is that the authors seem to have compared directly time series from observations and freely evolving climate models. The limitations associated with this issue should be treated with care and properly discussed.

Response: Thank your very much for your comments on the Method. We answer this question in more details in Q.1-6.

Q.1-2

Other possible applications of the method as suggested below should be considered in order to propose a more robust analysis of the method and its limitations.

Response: Thank your very much for your comments on the Method and its limitations. We answer this question in more details in Q.1-7 and Q.1-8.

Q.1-3

The introduction could be slightly improved by adding a couple more information about the context, some perspectives and a few more references.

Response: Thank your very much for your comments on Introduction. We answer this question in more details in Q.1-5.

Q.1-4

 The form of the text is understandable and the number of grammar errors limited although the paper could be improved by having it checked (once again ?)  by an English native speaker (which the reviewer is not).

Response: Thank you very much for the suggestion on improving the readability of the paper. We will make the manuscript proofread and improved by an English native speaker.

General comment about the introduction :

Q.1-5

Climate models biases spatio-temporal consistency or "stationarity" has important implication for the validity of climate model bias corrections methods (in this regards, see publication by Krinner and Flanner, 2018). This might be worth mentioning in the introduction.

Response: We will include the discussion on Climate models biases spatio-temporal consistency (Krinner and Flanner, 2018) in the Introduction.

General comment about the method and its application :

Q.1-6

1) It is unclear to me whether the matrix of biases (or residuals) for MIROC-5 and

FGOALS-g2 were obtained by directly comparing time series of the model to the monthly time series of GPCP data set or to the monthly climatology of GPCP? In any case, it is challenging or even questionable to directly compare the time series of freely evolving climate models to an observational time series due to the independence between the two time series as freely evolving climate models own their inner variability. Differences for a given month should therefore in any case not being called a "bias". Possibly, the fact that the bias detection algorithm is applied on a long climate period with thresholds on the size of the area and the length of the period to identify "bias families" allows for the method to identify actual biases of the climate model but this issue should be handled with care, more deeply explored and discussed. In my opinion, the application of the method proposed in this paper is more straightforward and easier to justify for the detection of biases in climate simulation (GCM or RCM) nudged towards meteorological reanalysis and directly compared to time series from observations (discussion of surface climate biases mostly in this case) or for the detection and analysis of climate anomalies as suggested by the authors in their discussion.

Response: The results shown in this manuscript is obtained by comparing the simulation outputs of climate models to observational time series of GPCP. The methodology for evaluating the effectiveness of the proposed clustering method in this paper is to check whether the bias patterns found by our algorithm are consistent with those found by existing studies. Towards this goal, in our case studies we intended to use the same climate model outputs and precipitation data as reported in the existing studies. Take the double ITCZ bias as an example. Both Oueslati et al. (2015, https://doi.org/10.1007/s00382-015-2468-6) and Siongco et al. (2015, https://doi.org/10.1007/s00382-014-2366-3) adopted GPCP for analyzing the double ITCZ bias in CMIP5 models. Adam et al. (2018, https://doi.org/10.1007/s00382-017-3909-1) performed their double ITCZ bias analysis using GPCP, the Climate Prediction Center (CPC) merged analysis precipitation (CMAP) product (Xie and Arkin 1996),

and the analyzed precipitation of the European Center for Medium-Range Weather Forecasts (ECMWF) Interim Reanalysis (Dee et al. 2011). However, only the results of GPCP were reported in their paper and others were omitted. Thus, we consider using GPCP in this case is proper. Note that it is very possible that meteorological reanalysis data are more suitable than observations for analyzing biases in other cases as suggested by the referee. Our proposed method works the same way for both meteorological reanalysis data and direct observations.

Q.1-7

2) Why focusing particularly on these two climate models and only on precipitation? Extending the analysis on other climate models would increase the robustness of the results and the conclusion drawn from them, while applying it to other climate variables (e.g. temperatures) would open possibilities for exploring links between biases among different variables (e.g. temperatures and precipitation) as the method is meant to be a tool for a better understanding of the sources of the climate model biases as the authors mentioned. If an extent of the application and the analysis on other climate models and/or other climate variables, the authors should at least justify why they restrained their analysis to these two climate models and to precipitation.

Response: The reason why we chose MIROC-5 and FGOALS-g2 is that there are a number of papers studying these two models, based on which we can check the effectiveness of the proposed approach. MIROC-5 and FGOALS-g2 are only used as case studies. We will modify the title of the paper (also as suggested by Q.2-3) and clarify the choice of models in the manuscript.

We did the experiments on temperatures and found meaningful bias patterns as well. We did not include the results mainly for space concerns. We will add this set of experiments to enhance the robustness of the proposed method in the manuscript.

Exploring the links between biases among different variables is indeed very useful. It requires measuring the correlations between bias patterns among different variables, which is our current undergoing research. We will discuss it more in the Summary and Discussion.

Q.1-8

3) Applied on monthly values as it is in this paper hampers the method from detecting model biases that evolve with the daily cycle (which is often the case for temperatures and precipitation). Would the method proposed in this paper be able to deal with this type of biases? If this is not explored by a short application of the method on data at a higher time resolution, this should at least be briefly mentioned and discussed.

Response: Our proposed clustering method is designed to find the bias patterns that steadily appear in a region over a period of time. Comparing to climate data, meteorological data (usually at a higher time resolution) is more dynamic, which cannot be handled by our proposed method. We will discuss it as a limitation of our method in the Summary and Discussion.

More particular, minor comments:

Q.1-9

Title : If the method is meant to become a useful tool for climate models analysis a more general and "attractive" title should be considered, for example "Spatio-temporal consistent bias pattern detection method : application on MIROC5 and FGOALS-g2 precipitation"

Response: The paper indeed focuses on a general clustering technique with MIROC5

and FGOALS-g2 used as case studies. We will modify the title to reflect the focus of this paper.

Q.1-10

P1 L18 : I would rather recommend the use of AOGCM or CGCM (used but nor defined in P2) acronym when speaking precisely of coupled atmosphere-ocean general circulation models.

Response: We will use the CGCM acronym for coupled atmosphere-ocean general circulation models throughout the manuscript.

Q.1-11

P1 L20 : May-be add a more general reference about biases and evaluation of climate models (e.g. Flato et al., 2013 from IPCC AR5).

Response: We will add the reference Flato et al.(2013) on P1 L20.

Q.1-12

P1 L22 : Since it is a widely spread and studied bias, may-be, add one or two more references about the double ITCZ.

Response: We will add more references about the double ITCZ on P1 L22.

P2 L30 : "Highlight" or "evidence" may be more appropriate here than "discover" ?

Response: We will change the sentence on P2 "...multivariate spatio-temporal clustering (MSTC) to discover recurring climate regimes from model outputs..." to "...multivariate spatio-temporal clustering (MSTC) to evidence recurring climate regimes from model outputs..."

**Q.1-13**

P3 L13-28 : A brief summary of the method (already present in the abstract) as well as of the results might not relevant at this point in the paper. At this point, readers might just want to know about what they will find in the different sections of the paper.

Response: We will rewrite this part to remove the repeated information (a brief summary of the method and results), and provide the organization of the paper and just the highlights our method.

**Q.1-14**

P4 L8 : Which statistical tests will be available for users to determine significance of the biases?

Response: The proposed method does not have any restrictions on which statistical tests should be chosen. Currently, we support the commonly used t-test.

**Q.1-15**

P7 L7 : How were the values of the parameters determined? Did you performed some sensitivity tests?

Response: We chose the values of the parameters by considering both the number and size of the resultant bias patterns. For example, $\delta_a$ defines the minimum number of grid points that must appear in a bias instance. A large $\delta_a$ value may filter out too many bias instances, whereas a small $\delta_a$ value may lead to tiny bias instances. For a given dataset, we tried a range of $\delta_a$ values and chose the one that traded off well. We will provide more guidelines on parameters in the README file.

Q.1-16

P8 L8 "Positive residuals are plotted in red and negative residuals are plotted in blue". I think this sentence should only appear in the caption of the figure.

Response: We will delete the sentence from L8 on P8 and only keep it in the caption of the figure.

Q.1-17

P9 L13 "In longer time series, the percentage will be even lower, which indicates that analysis or similarity calculations based on the entire series could be misleading" → I am not entirely sure about this statement, and it should in my opinion be explained more clearly if it is a meaningful result.

Response: Here we would like to emphasize that the discovered bias patterns are indeed local patterns temporally. If we take the global view of the entire time series, the discovered bias patterns only span a small portion of it and can be easily neglected under this circumstance. We will explain this result more clearly in the manuscript.

Q.1-18
P9L21 "One mainly consists of negative biases while the other mainly consists of positive biases. As a result . . .." This result might simply just be the consequence of the model and observational time series being completely independent and disconnected. As already mentioned, I think the authors should deal more carefully with this issue.

Response: It is true that the two opposite bias patterns in the specific region are simply coincidences. In the manuscript, we pointed out these two opposite bias patterns as an example of the patterns that could not be found if the entire time series had been considered. We will clarify this limitation on possible interpretations in the manuscript.

References :

Flato, G., Marotzke, J., Abiodun, B., Braconnot, P., Chou, S. C., Collins, W. J., Cox, P., Driouech, F., Emori, S., Eyring, V., et al.: Evaluation of Climate Models. In: Climate Change 2013: The Physical Science Basis. Contribution of Working Group I to the Fifth Assessment Report of the Intergovernmental Panel on Climate Change, Climate Change 2013, 5, 741–866, 2013.

Krinner, G. and Flanner, M.G., 2018. Striking stationarity of large-scale climate model bias patterns under strong climate change. Proceedings of the National Academy of Sciences, 115(38), pp.9462-9466. https://doi.org/10.1073/pnas.1807912115

---

## Author Comment (AC2) · 28 Sep 2019

Thank you for the efforts you have gone to supply the code and data your manuscript depends on. While you seem to have supplied the substance of what is required, the archiving and citation are currently not compliant with GMD policy.

Thank you for taking your time and effort in reviewing our manuscript and we appreciate your insightful comments for improving the manuscript.

**Q.1 Code on GitHub**

GitHub is a great development and distribution platform, but it's not a persistent archive. Even GitHub themselves tell you to use Zenodo for this! Please therefore provide a persistent archive of the exact version of the code that you use. The easiest way to do this is to use the Zenodo GitHub integration. See https://guides.github.com/activities/citable-code The resulting Zenodo repository will tell you how to cite the resulting DOI. You are welcome to also provide the reference to GitHub, for example as the preferred download location for the current version of the code, but this needs to be in addition to the DOI.

Response: Thank you for suggesting Zenodo for a persistent archive. We have published the code on Zenodo and the resulting DOI is 10.5281/zenodo.3456909. We will modify the reference in our manuscript accordingly.

Q.2

The comparison model data just points to the ESGF website. It is not clear which model data set was used. For CMIP5 ESGF provides unique data set IDs in the metadata. Please insert the relevant IDs in the code.

Response: The data sets used in this manuscript are 'cmip5.output1.MIROC.MIROC5.historical.mon.ocean.Amon.r1i1p1.v20131009|esgf-data1.diasjp.net' and 'cmip5.output1.LASGIAP.FGOALS-s2.historical.mon.ocean.Amon.r1i1p1.v20161204|esg.lasg.ac.cn' from the ESGF website. We will attach the data set IDs in our manuscript.

Q.3

For GPCP, the correct citation is given at https://data.nodc.noaa.gov/cgi-bin/iso?id= gov.noaa.ncdc:C00932#. Once again, it's perfectly OK to also include the URL, but

the citation is more permanent so should always also be used (you could also put the URL in the citation in the bibliography if you think that's neater).

Response: We will provide both the correct citation, Adler et al. (2011) as given in the GPCP website, and the URL for GPCP in the bibliography.

References :

Adler, R.F., G.J. Huffman, D. Bolvin (2011): Global Precipitation Climatology Project - Monthly, Version 2.2. NOAA National Climatic Data Center.

---

## Author Comment (AC3) · 28 Sep 2019

The manuscript presents the description of a method to cluster and classify main regions of bias in 3D grids and its application to the outputs of two climate models. The merit of the work is fair, improved by the fact that the code is shared and licensed under GPLv3. However, there are several shortcomings that need to be addressed.

Thank you for taking your time and effort in reviewing our manuscript and we appreciate your insightful comments for improving the manuscript.

Q.2-1

[Figure]

The proposed methodology is, someway, not new. Clustering analysis has been used to check model performance now for a long time and this is not properly put into context in the Introduction. It is surprising that the work by Yokoi et al. (2011) is not cited at all (https://doi.org/10.1175/2011JAMC2643.1) when moreover uses as examples MIROC5 and FGoals1. However, in the Introduction the authors cite other works that simply put the focus on regional studies or features, such as the ITCZ. I strongly recommend the authors to perform a better and more complete review of the literature on this topic and improve this section.

Response: Thank you for pointing out the relevant references that are currently missing in the Introduction. We will improve the Introduction by reviewing more existing clustering analysis works on model performance such as Yokoi et al. (2011) (https://doi.org/10.1175/2011JAMC2643.1), Sierra et al. (2018) (https://doi.org/10.1007/s00382-017-4010-5), and Khan et al. (2018) (https://doi.org/10.3390/w10121793), and clarify the differences between these works and ours. That is, the existing works such as Yokoi et al. (2011) focused on clustering performance metrics to group correlated metrics, whereas our proposed method is applied to climate model outputs directly and tries to find local spatio-temporal consistent bias patterns in the outputs.

Q.2-2

Also, the code (see also comment from the executive editor about using ZENODO) includes a README file, but this file does not contain any information about the structure of all the code in the repository. It should explain what each 'program' does and list all the files in the repository. Moreover, the different pieces of code that you share, do not have comments at all. Including commentaries in the code explaining what every set of routines or group of code does, is a basic request in software development and very important when you share it. Therefore, please, include

adequate commentaries in the software with explanations. Finally, in the text of the manuscript, when you mention the analysis and the tool to generate Figure 3, include the name of the code/routines that use to do it and make clear to the reader that they can be accessed from the link available in the 'Code and data availability section'.

Response: Thank you for your valuable suggestions on code readability. We will enhance our code readability by adding the structure of the code in the README file and adding adequate commentaries in our codes. Figure 3 in our manuscript was generated using routine 'plotGlobal' defined in 'plotHeatMap.py'. We will add the names of these routines in Section 2.3 and explain how a bias family can be visualized and analyzed by using our tool.

Other minor issues:

Q.2-3

- the current title of the manuscript is misleading. Readers can think that you perform a full detection and study of bias in the two models mentioned. Indeed, your manuscript is about a clustering technique and MIROC and FGOALS are simply used as testbeds without any interpretation. The title should reflect this, maybe something similar to 'An algorithm for spatio-temporal consistent bias pattern detection: a case study using MIROC5 and FGOALS-g2'

Response: It is indeed correct that MIROC and FGOALS were used as testbeds in this manuscript. We will modify the title to reflect that the focus of this manuscript is the proposed clustering technique and MIROC5 and FGOALS-g2 are used as case studies.

Q.2-4

- If you want to highlight the relevance of climate models for the study of climate change, probably the papers that you cite in the second line of the Introduction are not the most suitable. Better use Taylor et al. (2012) or IPCC reports.

Response: We will add the citation of Taylor et al. (2012) and IPCC reports to the second line of the Introduction.

Q.2-5

- When you mention the CMIP5, please, cite the paper by Taylor et al. (2012) (https://doi.org/10.1175/BAMS-D-11-00094.1).

Response: We will cite the reference of Taylor et al. (2012) on P1 when first mentioning the CMIP5.

Q.2-6

- The last sentence on page 2 is unnecessary, delete it.

Response: We will remove the last sentence on P2.

Q.2-7

- When you state on page 3 (first paragraph) that the computational cost would be too high, please, include extra detail at least on orders of magnitude of how more expensive it would be.

Response: We will explain in details why considering spatial and temporal characteristics simultaneously is of a high computational cost on P3. Actually, the computational cost of a naive joint clustering algorithm would be the product of the costs when focusing on spatial or temporal dimensions separately. In this paper, we mainly focus on bias patterns that steadily appear in a region over a period of time and thus we can reduce the cost by tracing the bias pattern along time. The time cost of the proposed method is affected by the distribution of underlying bias patterns. In terms of data used in the case study, it can give the analysis result in minutes.

Q.2-8

- when you mention the models used, please, provide links and cite the relevant papers with the description of the models.

Response: Links to download the data have been provided in 'Code and data availability' and we will attach detailed data set Ids for data downloaded from ESGF. In addition, we will cite the papers that describe the models themselves:

1. Bao, Qing, et al. "The flexible global ocean-atmosphere-land system model, spectral version 2: FGOALS-s2." Advances in Atmospheric Sciences 30.3 (2013): 561-576.

2. Watanabe, Masahiro, et al. "Improved climate simulation by MIROC5: mean states, variability, and climate sensitivity." Journal of Climate 23.23 (2010): 6312-6335.

The second paper has been cited in the manuscript and we will cite it the first time 'MIROC5' appears.

Q.2-9

- the citation and reference to HUANG Xin is capitalized, fix it.

Response: We will fix the capitalization of the reference to Huang Xin et al. (2019).

Q.2-10

- page 7: you have already explained what CMIP5 stands for in page 1, you do not need to repeat it here.

Response: We will use the abbreviation of CMIP5 directly on P7.